# The Emerging Role of Menstrual-Blood-Derived Stem Cells in Endometriosis

**DOI:** 10.3390/biomedicines11010039

**Published:** 2022-12-24

**Authors:** Mariana Robalo Cordeiro, Carlota Anjinho Carvalhos, Margarida Figueiredo-Dias

**Affiliations:** 1Faculty of Medicine, Gynecology University Clinic, University of Coimbra, 3000-548 Coimbra, Portugal; 2Gynecology Department, Coimbra Hospital and Universitary Center, 3004-561 Coimbra, Portugal; 3Coimbra Institute for Clinical and Biomedical Research (iCBR), Area of Environment, Genetics and Oncobiology (CIMAGO), Faculty of Medicine, University of Coimbra, 3001-301 Coimbra, Portugal

**Keywords:** menstrual-blood-derived stem cells, endometriosis, early diagnosis, pathophysiology

## Abstract

The human endometrium has a complex cellular composition that is capable of promoting cyclic regeneration, where endometrial stem cells play a critical role. Menstrual blood-derived stem cells (MenSC) were first discovered in 2007 and described as exhibiting mesenchymal stem cell properties, setting them in the spotlight for endometriosis research. The stem cell theory for endometriosis pathogenesis, supported by the consensual mechanism of retrograde menstruation, highlights the recognized importance that MenSC have gained by potentially being directly related to the genesis, development and maintenance of ectopic endometriotic lesions. Meanwhile, the differences observed between MenSC in patients with endometriosis and in healthy women underlines the applicability of these cells as a putative biomarker for the early diagnosis of endometriosis, as well as for the development of targeted therapies. It is expected that in the near future MenSC will have the potential to change the way we manage this complex disease, once their long-term safety and effectiveness are assessed.

## 1. Introduction

Endometriosis refers to endometrial tissue (glands and stroma) that is ectopically implanted in sites outside the uterine cavity. It mostly affects women of reproductive age, with an increasing incidence of up to 10–15% [1]. This complex disease is believed to result from a combination of several aberrant biological processes and mechanisms, including retrograde menstruation, lymphatic and vascular dissemination, metaplasia, immune dysfunction, genetic and epigenetic predisposition, presence of altered progenitor endometrial cells in the menstrual flow, and the existence of a favorable peritoneal microenvironment for endometriotic lesions [1,2,3].

Nevertheless, the exact etiology of endometrial ectopic tissue is still uncertain. Sampson’s theory is the most consensual explanation [2]. The concept underlying this theory highlights the importance of the menstrual fluid’s cellular components in the development of endometriosis. Among these cellular elements, mesenchymal stem cells (MSC) originating from the endometrium (eMSC) and menstrual-blood-derived stem cells (MenSC) have gained prominence by potentially being directly related to the genesis, development, and maintenance of ectopic lesions [3,4,5,6].

The differences observed between MenSC in patients with endometriosis and those in healthy women underlines the relevance in evaluating these cells, especially with respect to surface markers and gene expression levels, to better understand their emerging role in the occurrence of this disease [7,8,9].

This review summarizes the current knowledge and research regarding the potential value of menstrual-blood-derived stem cells in the pathogenesis, early diagnosis, and treatment of endometriosis.

## 2. Endometrial Stem Cells

Stem cells can be defined as a population of undifferentiated cells, usually arising from a single cell (clonal), characterized by the ability to extensively proliferate (self-renewal) and differentiate into different types of cells and tissues (potent). In this way, they are responsible for the development and regeneration of organ and tissue systems, as well as for the renewal of their populations and differentiatrion into multiple cell lineages [10].

The human endometrium refers to the mucosal lining of the uterus and comprises two regions with different structures and functions: the functionalis and the basalis. The functionalis consists of the upper two-thirds of the endometrium, whilst the basalis comprises the lower third [11,12,13]. In each menstrual cycle, and in the absence of pregnancy, the functionalis is shed during the menstrual phase as a result of hormonal changes, after which it is immediately reconstructed, while the basalis remains unchanged [13]. This cyclic proliferation and regeneration that occurs during each menstrual cycle was first proposed to be mediated by resident adult stem cell populations within the endometrium by Prianishnikov in 1978. Nearly 25 years later, the first evidence of stem cell activity in the human endometrium was reported, and clonogenic epithelial and stromal cells were identified. Over the years, several different populations of stem cells have been identified in the human endometrium [6,14,15]. 

Endometrial stem cells (EnSC) were initially thought to be located only in the basalis; however, robust evidence has confirmed the existence of stem cells in both the functionalis and basalis. This hypothesis has been supported by numerous studies using endometrial biopsy tissue showing the presence of small populations of adult endometrial stem cells with the classic stem cell properties of clonogenicity, self-renewal and differentiation [15]. In addition, the presence of EnSC in the basal layer of the endometrium was also successfully confirmed by their isolation from endometrial tissues after treatment with collagenase and from menstrual blood after density gradient centrifugation or direct treatment to lyse red blood cells [14].

Three kinds of stem cells exist in the human endometrium: epithelial stem cells, mesenchymal stem cells and endothelial stem cells, which are identified based on the expression of specific surface markers and on their differentiation potential [15]. 

CD140 + CD146b + eMSC show a perivascular localization in both the functionalis and basalis and have the potential to differentiate into different cellular lineages, including osteogenic, myogenic, adipogenic, and chondrogenic lineages. The CD29, CD44, CD73, CD90 and CD105 markers, which are collectively known as mesenchymal stem cell markers, are expressed in this population of cells, but endothelial and hematopoietic markers are not. CD140b + CD146 + eMSC express genes involved in angiogenesis, inflammation, immunomodulation and cellular crosstalk at high levels, together with the increased expression of multiple molecules involved in signaling pathways, including Notch, Hedgehog, IGF, TGFβ, and G protein-coupled receptor [16]. 

SUSD2 is considered a novel marker of eMSC and SUSD2+ cells predominantly reside in the perivascular region of the basalis and functionalis. These cells can differentiate into adipocytes, osteocytes, chondrocytes, myocytes, and endothelial cells, as well as yield endometrial stromal-like tissues. SUSD2+ cells express MSC markers including CD29, CD44, CD73, CD90, CD105, CD117, CD140b, CD146 and STRO-1 [16]. 

Side population (SP) cells are considered an adult stem cell marker and were first isolated from human endometrium in 2007 by Kato et al. These cells have the potential to differentiate into gland- and stromal-like cells, although the majority of endometrial SP cells are quiescent. They are present in the endothelial cells of both endometrial layers. SP cell-specific markers include CD31 (an endothelial cell marker); CD34 and CD54 (hematopoietic cell markers); EMA (an epithelial cell marker); and CD90, CD105, and CD146 (MSC markers) [16]. 

Endometrial epithelial stem cells were first isolated by Chan et al. in 2004. The specific markers identifying this cell population are still under investigation, where N-Cadherin and stage-specific embryonic antigen-1 (SSEA-1) have been suggested as two promising candidates by Cousins et al. Recently, it was also observed that multipotent endometrial epithelial stem cells with somatic mitochondrial DNA mutations in the CCO gene have the ability to regenerate endometrial glandular lineages [6,15,17].

## 3. Stem Cell Theory for Endometriosis Pathogenesis

Currently, the most widely accepted theory for endometriosis pathogenesis is endometrial ectopic implantation caused by retrograde menstruation, as proposed by Sampson in 1921. Sampson’s theory proposes that the endometrial glandular epithelial and mesenchymal cells enter the pelvic cavity by retrograde menstruation and are ectopically implanted in different sites, such as the ovary and the adjacent pelvic peritoneum, where they grow and spread to form ectopic lesions [2].

The initial stem cell theory for endometriosis pathogenesis proposes that endometriotic lesions arise from stem cells originating from neonatal or adult endometrium, bone marrow or derivates from Müllerian ducts. Regardless of their origin, it is believed that stem cells translocate into ectopic sites through fallopian-tube-mediated retrograde menstruation, lymphovascular circulation, and/or direct migration [3]. Consequently, they give rise to the various types of cellular components of the endometriotic implant, such as glandular, stromal, endothelial, and smooth muscle cells, through their differentiation, epithelial mesenchymal transition, and metaplasia [2,3] (Figure 1).

Although certain immunological imbalances have been reported to promote the development of endometriosis, the inherent nature of MenSC can also play a key role in the occurrence of this disease. Menstrual stage eMSC undergo more rounds of self-renewal than proliferative or secretory stage eMSC, which highlights their potential as lesion-initiating cells. In addition, due to their immunoregulatory functions, indoleamine 2,3-dioxygenase-1 (IDO1), cyclooxygenase-2 (COX-2) and forkhead transcription factor-3 (FOXP3) are considered as major candidates in fallowing MenSC to enter the peritoneal cavity [4,6,16].

## 4. Menstrual-Blood Derived Stem Cells

Endometrial stem cells can be obtained non-invasively from menstrual blood and are referred to as menstrual blood-derived stem cells. MenSC were first identified from menstrual blood in 2007 [16]. MenSC mainly express CD29, CD9, CD13, CD44, CD41a, CD73, CD59, CD90 and CD105 but not CD19, CD34, CD45, CD117, CD130 or human leukocyte antigen-DR isotype (HLA-DR). Interestingly, some studies have reported the positive expression of embryonic markers and intracellular multipotent markers, such as OCT-4, c-kit proto-oncogene (c-kit)/CD117 and stage-specific embryonic antigen-4 (SSEA-4), which do not exist in MSCs from other sources. However, controversies exist regarding positivity for c-kit and SSEA-4 [8,16,17,18].

Recent studies have reported that MenSC can increase by one doubling every 20 h when supplied with sufficient culture conditions, which is nearly twice as fast as the doubling time of bone marrow MSC (BM-MSC). Differences in telomerase activity may partially explain this highly proliferative characteristic. The overexpression of embryonic trophic factor and extracellular matrix observed in MenSC is influenced by the characteristic high proliferative rate of these cells, which, alongside their stable genetic characteristics and apparent pluripotency, suggest that these novel stem cells may have unexpected therapeutic properties. MenSC also have an extraordinary broad differentiation capacity. These cells can differentiate into adipocytic, osteogenic, cardiomyocytic and neuronal lineages, as well as respiratory epithelial, endothelial, myocytic, hepatic, germ-like and pancreatic cells. Liu et al. also showed that MenSC can differentiate into ovarian tissue-like cells [16,17,18,19].

MenSC interact with a wide variety of immune cells and participate in the regulation of cellular and humoral immunities. The expression of MHC-I by MenSC also indicate their capability for immunomodulation. In addition, in vivo studies have shown that MenSC may protect the liver from acute injury in mice through their anti-inflammatory and immunomodulatory effects [8,16,17]. Similar to SUSD2 + eMSC, MenSC inhibit the optimal phenotypic differentiation of human peripheral blood monocytes (PBMC) into immature and mature dendritic cells (DC in which IL-6 and IL-10 are considered essential [16].

Moreover, menstrual blood is not only a source of MenSC but also DC, which are professional antigen-presenting cells that form an essential interface between the innate sensing of pathogens and the activation of adaptive immunity [16,17].

## 5. MenSC in Patients with Endometriosis vs. Patients without Endometriosis

MenSC from patients with endometriosis (E-MenSC) display different morphologic, phenotypic, and functional characteristics when compared to MenSC from women without endometriosis (NE-MenSC).

With respect to morphology, NE-MenSC have a characteristic fibroblast-like spindle shape, whereas E-MenSC are described as being less stretched and elongated. Sahraei et al., through treatment with a conditioned medium derived from NE-MenSC, observed morphologic changes in E-MenSC, which acquired a more elongated, spindle-shaped morphology. In addition, as shown by Nikoo et al., E-MenSC form small colonies when grown in 3D cultures, which are absent in NE-MenSC cultures [7,8,16,19].

Concerning the differences in CD marker expression profiles, both E-MenSC and NE-MenSC were considered positive for CD9, CD10 and CD29 by Nikoo et al., although their expression was significantly higher in E-MenSC that in NE-MenSC. In contrast, Sahraei et al. found that CD9 and CD29 expression was significantly lower in E-MenSC, while CD10 was highly expressed. Moreover, after treatment with NE-MenSC-derived conditioned medium, CD10 expression decreased significantly in E-MenSCs, providing the CD10 surface marker with an important role in the diagnosis of endometriosis [16,20].

E-MenSC have been shown to possess higher proliferation and invasion capacities when compared with NE-MenSC. However, no differences were observed regarding their adhesion ability in Sahraei et al.’s research. Nevertheless, it is worth noting Nikoo et al.’s findings of higher CD29 expression levels in E-MenSC, which suggest that these cells might support attachment to the peritoneum. On the other hand, E-MenSC have shown a higher migratory capacity, shown by higher mRNA expression levels of the MMP-2 and MMP-9 genes in these cells when compared to NE-MenSC [7,8,21,22].

The apoptosis rate in the endometrial cells of women with endometriosis is decreased, suggesting that the survival rate of cells that reach the peritoneal cavity is higher in patients with progressive endometriosis. Sahraei et al.’s results suggest that the Bax/Bcl-2 ratio is significantly lower in E-MenSC than in NE-MenSC, which has been shown to have an important association with apoptosis, thus confirming the decreased apoptosis seen in endometriosis stromal cells [7,8,21,22].

Angiogenesis plays an essential role in the establishment and growth of endometriotic lesions, regardless of apoptosis. Likewise, vascular endothelial growth factor (VEGF), considered as one of the most active angiogenic factors, is highly involved in both physiological and pathological angiogenesis. In endometriotic implants, growth factors, hormones, cytokines, and hypoxia stimulate the production of VEGF, which is secreted by the ectopic endometrium and peritoneal macrophages. In accordance, Sahraei et al. observed a high level of VEGF expression in E-MenSC [7,8,21,22].

There is accumulating evidence suggesting that a set of immune response-related malfunctions plays a key role in the development and progression of endometriosis. E-MenSC are considered to have greater potential in directing inflammatory responses. Regarding immunomodulation functions, Nikoo et al. first demonstrated that E-MenSC were different from NE-MenSC, exhibiting a higher expression of enzymes such as IDO1 and COX-2, while higher levels of FOXP3—a transcription factor involved in the development and function of regulatory T-cells—were found in E-MenSC which is. In addition, Sahraei et al. provided evidence that E-MenSC exhibit increased expression levels of proinflammatory cytokines and chemokines such as IL-1b, IL-6, and IL-8, secreted by peritoneal macrophages and ectopic endometriotic lesions, and the decreased expression of TNF-α, which might be another factor contributing to a proinflammatory peritoneal microenvironment. In addition, the expression of the transcription factor NF-κβis also higher in E-MenSC, promoting inflammation, invasion, angiogenesis, and cell proliferation in endometriotic lesions, as well as in the peritoneal macrophages of patients with endometriosis [7,8,21,22].

In endometriotic lesions, the inherent inflammation and estrogen exposure form a positive feedback loop, increasing the expression of aromatase and COX-2 and local estrogen production. Hence, it comes as no surprise that the expression of the estrogen gene in E-MenSC was found to be significantly higher than that in NE-MenSC [7,8,21,22].

There has been an overall attempt to understand the role of stemness-related genes in endometriosis. The increased expression of stemness-related markers in endometriotic tissue suggests that it can promote cell survival and self-renewal. A higher mRNA expression of NANOG, OCT-4, and SOX-2 genes, as well as miR-200b-3p upregulation, have also been observed in E-MenSC [7,8,21,22].

Penariol et al. were the first to apply multi-omics approaches in MenSC evaluation [9]. Among the differentially expressed genes and proteins studied, the genes ATF3, ID1, ID3, FOSB, SNAI1, NR4A1, EGR1, LAMC3, and ZFP36 and the proteins COL1A1, COL6A2, and NID2 were considered to be overexpressed in E-MenSC when compared to NE-MenSC, unlike the downregulation of the MT2A and TYMP proteins seen in E-MenSC. These findings are consistent with the previously mentioned MenSC characteristics that result in the increased cell proliferation, stemness, and accentuated mesenchymal–epithelial transition process observed in endometriosis [9] (Table 1).

## 6. MenSC-Based Early Diagnosis

Patients with endometriosis often experience a delay from the onset of symptoms to a definitive diagnosis, which can be up to 11 years. Consequently, a delay in treatment also occurs, potentially resulting in disease progression and increased severity. Based on the previously reported evidence in the literature, it is clear that MenSC are key to the early diagnosis of endometriosis in the future, with an enormous potential to shift this paradigm [7,8,17,21,22].

As mentioned earlier, MenSC from patients with endometriosis differ from those of patients without endometriosis with respect to their morphology, phenotype, and mechanisms of action. Taking all this together, in combination with the fact that menstrual blood is an easily obtained, renewable, inexpensive, and non-invasive source of MenSC, these cells could be used as promising targets for the early diagnosis of endometriosis [7,8,17,21,22].

These findings pave the way for novel multi-omics approaches to evaluate differences in MenSC between healthy individuals and patients with endometriosis. Therefore, it will be possible to optimize the combination of potential biomarkers, especially genes and proteins, and develop non-invasive diagnostic strategies. On the other hand, MenSC might also be helpful in predicting the risk of developing endometriosis in the future for healthy women [7,8,17,21,22].

Another interesting approach to the early diagnosis of endometriosis was suggested by Cousins et al. and is related to neonatal uterine bleeding (NUB) (Figure 2).

At birth, neonates suffer from menstrual bleeding due to the shedding of the decidual endometrium following the sudden decline in maternal hormones [6]. Their relatively long cervix, being functionally blocked with thick mucus, suggests that retrograde NUB may occur. It is hypothesized that fetal endometrial stem cells have the potential to gain access to the neonatal pelvic cavity, invading the mesothelium and remaining dormant until the rise of estrogen levels at puberty [6].

## 7. MenSC-Based Therapies

The appeal of MenSC-based therapies mainly relates to the regular and non-invasive way that these cells can be obtained from menstrual blood. There is an undeniable therapeutic potential in MenSC as they exhibit the ability to migrate into injury sites, differentiate into distinct cell lineages, secrete soluble factors, and regulate immune responses, much like bone marrow mesenchymal stem cells [16,17].

As previously mentioned, their high proliferation rate allied with the stable genetic characteristics of MenSC, as well as their apparent pluripotency, make these cells promising candidates in stem cell therapy for inflammation and immune-related diseases. The aforementioned capacity of MenSC to differentiate into various cell types, confirms that these stem cells may exhibit unexpected therapeutic properties in the treatment of a variety of diseases in regenerative medicine. In fact, the therapeutic potential of MenSC has already been established for various diseases in pre-clinical research. Despite all these possible applications, MenSC could play a key role in the treatment of endometriosis given their major involvement in its pathogenesis [16,17].

By comparing the morphology, expression of surface markers, cell proliferation, invasion, adhesion, and immunomodulatory abilities of E-MenSC and NE-MenSC, the results of Sahraei et al. and Nikoo et al. clearly highlight the presumed role of MenSC in improving endometriosis. In addition, the increased expression of IDO1, COX-2, IFN-γ, IL-10, and MCP-1 and the decreased levels of FOXP3 observed in the co-culture of E-MenSC and peripheral blood mononuclear cells also supports this therapeutical potential [4,8,9,14,17].

Regarding patient safety, the procedures for menstrual blood sample collection and MenSC isolation must be performed under aseptic conditions in agreement with the good manufacturing practice standards.

The evidence presented in this review needs to be carefully considered for clinical practice since in vitro results are not easy to extrapolate to in vivo systems, mainly because the masking of the cellular environment might occur. Therefore, it is crucial to establish ideal molecular markers for MenSC with high quality and consistency so that clinical trials can be performed and the standardization and validation of these cells in endometriosis can become a novel therapeutic weapon.

## 8. Conclusions

The pathogenesis of endometriosis is still unclear; however, there is evidence suggesting that endometrial stem cells, once in endometriotic lesions, can differentiate into endometrial-like tissue and contribute to the genesis and progression of endometriotic implants. Hence, MenSC are attractive cells to study mainly due to their proliferation, differentiation, and immunomodulatory characteristics, in association with the regular and non-invasive way they can be obtained. Furthermore, the differences found between MenSC from patients with endometriosis and those without endometriosis pave the way to the identification of biomarkers for early diagnosis and the development of targeted therapies. This potential non-invasive diagnostic tool and medical therapy would also help in improving patients’ quality of life, avoiding the unpleasant complications of endometriosis surgery. Once the long-term safety and effectiveness of MenSC in endometriosis treatment are assessed, these cells will have the relevant potential to change the way we manage this complex disease.

## Figures and Tables

**Figure 1 biomedicines-11-00039-f001:**
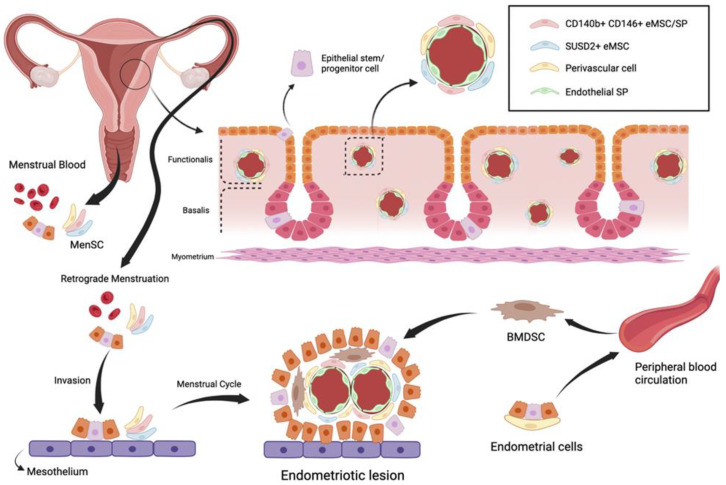
Stem cells are predominantly located in the basal layer of the endometrium. CD140b + CD146 + eMSC and SUSD2+ eMSC are located perivascularly and side population (SP) cells are located in the vascular endothelium, in both the basalis and functionalis layers of the endometrium. According to the retrograde menstruation theory, menstrual blood, containing MenSC and epithelial stem cells, flows toward the fallopian tubes and enters the pelvic cavity, invading the mesothelium and generating endometriotic implants. Endometriotic cells secrete cytokines that recruit bone marrow derived-stem cells (BMDSC) and incorporate them into the lesions. Endometriotic cells also enter the bloodstream causing distant implants [15].

**Figure 2 biomedicines-11-00039-f002:**
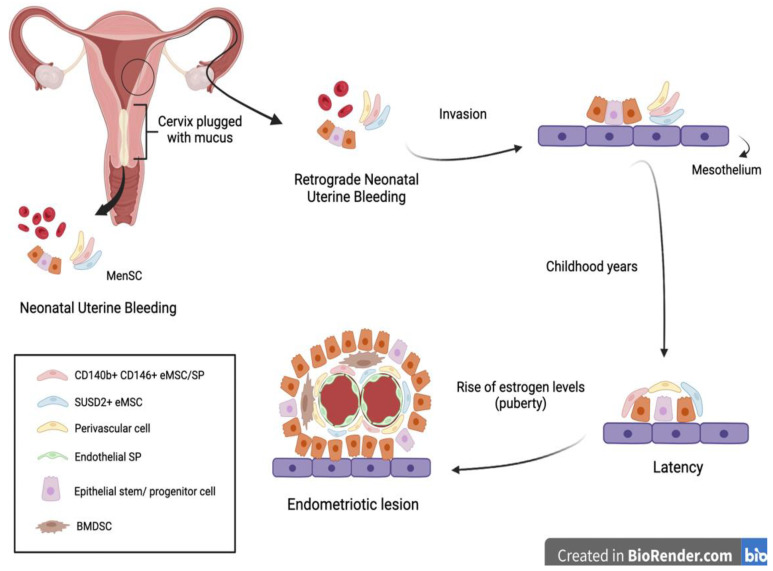
In some neonates, the withdrawal of maternal progesterone causes a shedding of the endometrium causing neonatal uterine bleeding. Similar to menstruation, this blood contains epithelial stem cells and MenSC. The passage of these fragments through the vagina can be obstructed by the presence of a cervix plugged with mucus. Therefore, it is hypothesized that the fragments follow a retrograde route through the fallopian tubes and invade the mesothelium. There, they remain latent during childhood until the rise of estrogen levels during puberty, at which point they are activated and form an endometriotic lesion [6]. BMDSC—bone marrow-derived stem cells; SP— side population.

**Table 1 biomedicines-11-00039-t001:** Differences in MenSC observed between patients without endometriosis vs. patients with endometriosis.

Characteristics	NE-MenSC	E-MenSC	References
Morphology	Fibroblast-like spindle shape	Less stretched and elongated	[8,20]
**Surface Markers Expression**			
CD9	+	+/−	[8,20]
CD10	+	++	[8,20]
CD29	+	+/−	[8,20]
Proliferation capacity	+	++	[8]
Invasive capacity	+	++	[8]
Adhesion capacity	+	+	[7,8]
**Angiogenesis**			
VEGF	+	++	[7,8]
**Migratory capacity**			
*MMP-2*	+	++	[7,8,21,22]
MMP-9	+	++	[7,8,21,22]
Apoptosis	++	+	[7,8]
Bax/Bcl-2 ratio	++	+	[7]
**Immunomodulation**			
IDO1	+	++	[8]
COX-2	+	++	[7,8,15,20]
FOXP3	++	+	[8]
IL-1β	+	++	[7,15,20]
TNF-α	++	+	[20]
IL-6	+	++	[7,15]
IL-8	+	++	[7,15]
NF-κβ	+	++	[7]
**Stemness-related genes**			
NANOG	+	++	[7,8,21,22]
OCT-4	+	++	[7,8,21,22]
SOX-2	+	++	[7,8,21,22]
miR-200b-3p	+	++	[22]
**Differentially expressed genes**			
ATF3	+	++	[9]
ID1	+	++	[9]
ID3	+	++	[9]
FOSB	+	++	[9]
SNAI1	+	++	[9]
NR4A1	+	++	[9]
EGR1	+	++	[9]
LAMC3	+	++	[9]
ZFP36	+	++	[9]
**Differentially expressed proteins**			
COL1A1	+	++	[9]
COL6A2	+	++	[9]
NID2	+	++	[9]
MT2A	++	+	[9]
TYMP	++	+	[9]

++ Reported as higher expression level. + Reported as lower expression level. +/− Reported as inconsistent level of expression between authors.

## Data Availability

Not applicable.

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
