# Peer review of "The Emerging Role of Menstrual-Blood-Derived Stem Cells in Endometriosis"

_biomedicines, 2022, doi:10.3390/biomedicines11010039_

Round 1
Reviewer 1 Report
dear authors
I congratulate with you for the well written qualitative systematic review on a new emerging topic.
I would underline the importance of non invasive diagnosis and medical therapies since surgery is associated to unpleasent complications (i.e. 10.1111/ajo.12358).
Reviewer 2 Report
This is an interesting and well written review on the potential role of menstrual blood-derived stem cells (MenSC) in endometriosis. The phenotype of endometrial stem cells and of MenSC has been described. The theory involving stem cells in the pathogenesis of endometriosis has been also revised and the differences between MenSC derived from women with and without endometriosis have been listed. The paper provides some contribution to the current literature.
The only concern refers to the addition of too many speculations in the text. The authors should refrain from introducing aspects yet to be demonstrated and for which other explanations are possible. For instance:
Line 223: ‘enabling the establishment of a proinflammatory peritoneal microenvironment’. It is totally possible that it is the inflammatory environment that favours the activation of the cells toward an inflammatory
Paragraph on early diagnosis: this paragraph is too much emphasized. The Authors should remember that the differences between eutopic and ectopic endometrium have not resulted in any biomarker so far. The rationale to foresee something different in menstrual cells is not supported by data.
Line 282: ‘we can consider NUB as a putative biomarker for the development of future endometriosis and a sign for early intervention to prevent disease progression’. Only speculation.
Overall, the authors should do some effort to limit their review to the evidence and to avoid speculations all over the text. These speculations are likely to address the opinions and the theory toward some research areas that so far have poor scientific grounds and this is dangerous.
